# Incidence Rate and Determinants of Recurrent Cholesteatoma Following Surgical Management: A Systematic Review, Subgroup, and Meta-Regression Analysis

**DOI:** 10.3390/biomedicines13102506

**Published:** 2025-10-14

**Authors:** Saqr Massoud, Raed Farhat, Uday Abd Elhadi, Rifat Awawde, Shlomo Merchavy, Alaa Safia

**Affiliations:** 1Department of Otolaryngology, Otology Unit, Ziv Medical Center, Safed 1311001, Israel; 2Department of Internal Medicine, Emek Medical Center, Afula 1834111, Israel

**Keywords:** lesteatoma, recurrence, tympanomastoidectomy, canal wall up, canal wall down, mastoid obliteration, second-look surgery, meta-analysis, meta-regression

## Abstract

**Background/Objectives:** Cholesteatoma is a destructive middle ear pathology that can cause chronic infection, ossicular erosion, and hearing loss. While surgical excision is the standard treatment, recurrence remains a major clinical challenge, and comprehensive data on long-term outcomes are limited. This meta-analysis evaluated cholesteatoma recurrence rates following surgery, identified clinical and surgical predictors of recurrence, and assessed trends across follow-up durations, techniques, and patient demographics. **Methods:** We searched PubMed, Scopus, Web of Science, CENTRAL, and Google Scholar for relevant studies (CRD42024550351). Studies reporting postoperative recurrence were included. Data on demographics, surgical approach, cholesteatoma type, and outcomes were extracted. Risk of bias was assessed using the Newcastle–Ottawa Scale. Pooled recurrence rates were calculated using random-effects models, and subgroup and meta-regression analyses were performed to identify predictors. **Results:** Eighty-four studies comprising 12,819 patients were included. The cholesteatoma recurrence rate showed geographic variability. Recurrence was higher in children (13%) than adults (10%), and in acquired (12%) versus congenital (7%) cholesteatoma. Advanced-stage disease, left-sided lesions, and revision surgeries increased recurrence risk. Canal wall down had lower recurrence (7%) than canal wall up techniques (16%). Adjuncts such as mastoid obliteration, ossicular reconstruction, and planned second-look surgeries reduced recurrence. Cumulative recurrence reached 39% at 15 years and 33% at 25 years. Meta-regression identified age, staged procedures, and second-look surgeries as independent predictors. **Conclusions:** Cholesteatoma recurrence is influenced by age, surgical approach, and disease severity. CWD procedures and comprehensive surgical planning reduce recurrence risk. Long-term follow-up and standardized outcome definitions are essential to improve monitoring and disease control.

## 1. Introduction

Cholesteatoma is a destructive lesion of the middle ear and mastoid characterized by the proliferation of keratinizing squamous epithelium with the potential for progressive bone erosion and serious complications, including hearing loss, facial nerve paralysis, and intracranial infections [1]. It is classically categorized as either congenital or acquired, the latter being far more prevalent and often associated with eustachian tube dysfunction, tympanic membrane retraction, or chronic otitis media [2]. Despite advances in surgical techniques, recurrence of cholesteatoma remains a persistent clinical challenge, with reported rates varying widely across studies—from as low as 3% to as high as 30%—depending on patient demographics, surgical strategy, and duration of follow-up [3,4,5].

Surgical eradication remains the mainstay of treatment. The two principal approaches—canal wall up (CWU) and canal wall down (CWD) tympanomastoidectomy—are selected based on disease extent, patient age, and surgeon preference [2]. While CWU techniques aim to preserve anatomical integrity and improve postoperative quality of life, they have been historically associated with higher rates of residual and recurrent disease [6]. In contrast, CWD procedures offer better disease control but may compromise long-term functional and esthetic outcomes. To balance these competing priorities, additional techniques such as mastoid obliteration, staged surgery, ossicular reconstruction, and endoscopic approaches have been introduced in recent years [6,7].

Several systematic reviews and meta-analyses [3,8,9,10,11] have attempted to address specific aspects of recurrence in cholesteatoma—whether by comparing CWU and CWD techniques in pediatric populations, evaluating surgical outcomes by approach, or exploring the utility of imaging modalities like non–echo-planar diffusion-weighted MRI in detecting residual disease [12,13]. However, to date, there has been no comprehensive synthesis incorporating both clinical and surgical predictors of recurrence across patient subgroups, operative strategies, and long-term follow-up intervals.

The present systematic review and meta-analysis aim to fill this critical gap by providing an in-depth, global evaluation of recurrence rates in cholesteatoma surgery. Specifically, we (1) quantify recurrence rates across countries and patient populations, (2) explore the impact of cholesteatoma type, stage, and location, (3) assess the comparative effectiveness of surgical techniques, including canal wall management, mastoid obliteration, ossicular reconstruction, and endoscopic approaches, and (4) examine time-dependent trends in recurrence through cumulative follow-up analysis.

## 2. Materials and Methods

This review was registered on PROSPERO (registration number: CRD42024550351). This research was carried out according to PRISMA guidelines [14]. The study protocol was prospectively registered, and all methodological decisions were made a priori to minimize bias. Artificial intelligence (AI) was not used in the conduct of any of the research steps; however, it was used in the writing of some parts, which were validated and edited by the researchers.

### 2.1. Search Strategy

A comprehensive literature search was performed across five electronic databases: PubMed, Scopus, Web of Science, CENTRAL, with the help of Google Scholar (only the first 200 were retrieved, as per recent recommendations) [15]. The search was conducted on 19 May 2024, and was restricted to studies published from the year 2000 onward. The search strategy for PubMed is detailed in (Appendix A), utilizing both Medical Subject Headings (MeSH) and text words related to cholesteatoma, recurrence, incidence, risk factors, and surgical interventions. Equivalent search strategies were adapted for the other databases. Studies were restricted to those published in English language.

### 2.2. Eligibility Criteria

Studies were included if they met the following criteria: (1) observational studies (cohort, case–control, or cross-sectional) or randomized controlled trials (RCTs) evaluating recurrence rates of cholesteatoma following surgical intervention, (2) reporting on at least one predefined outcome of interest, including recurrence rate and associated risk factors, (3) original reports including at least 20 patients, and (4) studies with extractable data suitable for meta-analysis. Exclusion criteria encompassed case reports, narrative reviews, editorials, conference abstracts, and studies without sufficient statistical data. We also excluded articles primarily focused on recidivism and residual disease rates instead of recurrence. Additionally, we ruled out all studies focusing on the diagnostic accuracy of various imaging techniques in detecting cholesteatoma recurrence (reporting sensitivity, specificity, negative/positive predictive values).

### 2.3. Study Selection and Data Extraction

Retrieved studies were screened based on titles and abstracts, followed by a full-text review to determine final inclusion. Any discrepancies, if found, were resolved by consensus or by consulting a third reviewer. Data extraction was performed using a standardized form, collecting information on study characteristics, patient demographics, surgical techniques, follow-up duration, and recurrence outcomes. Study-level characteristics included study design, country, year of investigation, sample size, and follow-up period. Patients’ data included cholesteatoma-related data (i.e., type, location, etc.), surgical intent (primary vs. revision surgery), staged surgery (single or multiple), planned 2nd look surgery, age group (children < 18 years vs. adults > 18 years), ossicular erosion and reconstruction, as well as detailed on the performed surgery (i.e., canal wall up/down, endoscopic approach, etc.).

### 2.4. Risk of Bias and Quality Assessment

The risk of bias in included studies was assessed using the Newcastle–Ottawa Scale (NOS) for observational studies [16]. Each study was evaluated for selection bias, comparability, and outcome assessment. An overall grade of poor, fair, and good methodology was given.

### 2.5. Statistical Analysis

A meta-analysis was performed using a random-effects model to account for anticipated heterogeneity across studies. Due to the considerable variability in patient populations, disease characteristics, and surgical techniques, providing a single pooled overall estimate was deemed inappropriate. Instead, we conducted extensive subgroup analyses to explore recurrence patterns based on patient demographics, cholesteatoma-related features, and surgical variables. Cumulative recurrence rates were also stratified by follow-up duration. Heterogeneity was quantified using the I-squared (I^2^) statistic, with values > 50% indicating moderate-to-high heterogeneity [17].

When overlapping cohorts were identified across publications, we included only the study with the largest sample size and longest follow-up to avoid double-counting. For subgroup analyses, we ensured that no cohort contributed more than once to the same subgroup estimate; thus, each subgroup rate was derived from independent study samples.

To better understand the sources of heterogeneity, we performed meta-regression analyses using continuous variables reflecting the proportion (%) of patients exposed to each surgical or clinical factor (e.g., percentage undergoing revision surgery or planned second-look procedures), rather than dichotomizing exposure (i.e., 0% vs. 100%). Publication bias was evaluated through visual inspection of funnel plots and Egger’s regression test. All analyses were conducted using Stata version 18 (StataCorp, College Station, TX, USA).

## 3. Results

### 3.1. Literature Search Results

A total of 2506 records were initially identified. After removing 985 duplicate records using EndNote, 1521 unique records were screened based on titles and abstracts (Appendix A). Full texts of 365 studies were sought for retrieval, with 8 reports not retrievable. Subsequently, 357 full-text articles were assessed for eligibility. A total of 273 articles were excluded for the following reasons: lack of recurrence data (*n* = 148), non-English language (*n* = 10), inclusion of only recurrent cases at baseline (*n* = 3), review articles (*n* = 2), duplicate publications (*n* = 7), book chapters (*n* = 1), small sample size (<20 cases; *n* = 7), diagnostic accuracy studies (*n* = 24), abstract-only publications (*n* = 7), and studies published before the year 2000 (*n* = 64). Ultimately, 84 studies met the eligibility criteria (Figure 1) [18,19,20,21,22,23,24,25,26,27,28,29,30,31,32,33,34,35,36,37,38,39,40,41,42,43,44,45,46,47,48,49,50,51,52,53,54,55,56,57,58,59,60,61,62,63,64,65,66,67,68,69,70,71,72,73,74,75,76,77,78,79,80,81,82,83,84,85,86,87,88,89,90,91,92,93,94,95,96,97,98,99,100,101].

### 3.2. Characteristics of Included Studies

The baseline characteristics of included studies are summarized in Appendix A. Most evidence was retrieved from Japan and Italy (12 studies each). A total of 12,819 cholesteatoma patients were included, of whom 4642 were males and 3185 were females. The gender of remaining patients was not disclosed. The follow-up period ranged from 6 to 300 months. In terms of surgery, CWU procedures were performed in 27 studies, CWD procedures were conducted in 29 studies, endoscopic surgery was conducted in 14 studies, and intact canal wall procedures were conducted in 8 studies. Recurrent cholesteatoma was defined in 31 studies, while the remaining 53 did not provide a definition criterion for it (Appendix A).

### 3.3. Methodological Quality of Included Studies

As assessed by the NOS, only 3 studies had good quality, 4 had fair quality, and 77 (91.67%) of studies had poor quality (Appendix A). Poor quality was due to multiple factors, including lack of confounding control (either by matching at baseline or regression models), improper assessment of outcome (lack of definition criteria for cholesteatoma recurrence), and lack of reporting of dropout (non-response) rate.

### 3.4. Incidence Rate of Cholesteatoma Recurrence

Table 1 shows the collative recurrence rate of cholesteatoma stratified by examined patients’ clinicodemographic characteristics.

#### 3.4.1. Country-Based Recurrence Rates

The recurrence rate of cholesteatoma varied across different countries (Appendix A). Among studies from China (4 studies, 3%; 95% CI: 1–6%, I^2^ = 0.05%), England (1 study, 3%; 95% CI: 0–6%), and Belgium (1 study, 3%; 95% CI: 0–8%), the recurrence rates were among the lowest. In contrast, the highest pooled recurrence rate was observed in India (3 studies, 29%; 95% CI: 0–70%, I^2^ = 98.61%), followed by Denmark (3 studies, 24%; 95% CI: 0–48%, I^2^ = 98.91%) and Israel (2 studies, 24%; 95% CI: 15–34%, I^2^ = 10.88%).

#### 3.4.2. Age-, Gender-, and Laterality-Based Recurrence Rate

The recurrence rate was higher in children (13%; 36 studies) compared with adults (10%; 25 studies). This difference remained consistent across multiple analyses, underscoring the prognostic role of patient age group. Studies that included both adults and children reported an intermediate recurrence rate of 11% (15 studies; 95% CI: 4–18%, I^2^ = 97.19%) (Appendix A).

The pooled recurrence rates were comparable between males (3 studies; 17%; 95% CI: 8–27%, I^2^ = 76.09%) and females (3 studies; 18%; 95% CI: 10–25%, I^2^ = 24.20%). While heterogeneity was moderate among male patients (I^2^ = 76.09%), it was considerably lower among female patients (I^2^ = 24.20%), indicating relatively consistent findings across studies in the latter group (Appendix A).

The recurrence rate was higher for cholesteatoma affecting the left ear (3 studies; 18%; 95% CI: 3–32%, I^2^ = 77.29%) compared to the right ear (3 studies; 15%; 95% CI: 6–25%, I^2^ = 31.87%). Heterogeneity was substantially greater in the left-ear subgroup (Appendix A).

#### 3.4.3. Recurrence Rate Based on Cholesteatoma-Related Characteristics

In terms of type, the recurrence rate was higher in acquired cholesteatoma (14 studies; 12%; 95% CI: 8–16%; I^2^ = 80.35%) compared to congenital cholesteatoma (13 studies; 7%; 95% CI: 3–11%; I^2^ = 92.36%) (Appendix A).

Regarding stage, the recurrence rate increased with disease severity. Stage I cholesteatoma had the lowest recurrence rate (8 studies; 4%; 95% CI: 0–7%; I^2^ = 33.70%), whereas Stage IV cholesteatoma exhibited the highest (5 studies; 18%; 95% CI: 9–28%; I^2^ = 0%). Intermediate stages showed a gradual increase in recurrence: Stage II (10%) and Stage III (14%), with higher heterogeneity observed in the latter stages (Figure 2).

In terms of location, the highest recurrence rates were reported for aural cholesteatoma (2 studies; 20%; 95% CI: 12–28%; I^2^ = 86.74%) and petrous bone cholesteatoma (2 studies; 19%; 95% CI: 5–33%; I^2^ = 74.96%). Recurrence was also notable in mastoid cholesteatoma (3 studies; 16%; 95% CI: 3–30%; I^2^ = 72.51%) and middle ear cholesteatoma (not specified) (7 studies; 16%; 95% CI: 9–23%; I^2^ = 86.57%). Lower recurrence rates were observed in pars flaccida (12%), pars tensa (8%), and tympano-mastoid cholesteatoma (1%); the latter showed minimal heterogeneity (I^2^ = 14.48%) (Appendix A).

In terms of definition, studies that provided a clear definition of recurrence reported a pooled recurrence rate of 11% (31 studies; 95% CI: 7–14%; I^2^ = 96.61%), while those that did not define recurrence showed a slightly lower rate of 10% (53 studies; 95% CI: 7–12%; I^2^ = 97.64%). The high heterogeneity (I^2^ > 96%) in both groups highlights inconsistencies in the criteria used for defining recurrence across studies (Appendix A).

#### 3.4.4. Surgical Intent-Based Recurrence Rates

The recurrence rate was lower in primary surgery cases (8 studies; 9%; 95% CI: 5–14%; I^2^ = 84.59%) compared to revision surgery (12 studies; 17%; 95% CI: 7–27%; I^2^ = 95.39%) (Figure 3).

Single-stage surgery had a recurrence rate of 8% (11 studies; 95% CI: 4–13%; I^2^ = 91.41%). In contrast, staged surgery (≥2 stages) showed a slightly higher recurrence rate of 9% (5 studies; 95% CI: 6–12%) with minimal heterogeneity (I^2^ = 0.05%) (Appendix A).

Studies that planned a second-look surgery showed a recurrence rate of 13% (4 studies; 95% CI: 6–19%; I^2^ = 60.80%), whereas those where a second look was not planned had a recurrence rate of 10% (7 studies; 95% CI: 7–12%; I^2^ = 32.64%) (Appendix A).

#### 3.4.5. Surgical Technique-Based Recurrence Rates

Appendix A shows the surgery-specific recurrence rate of cholesteatoma. CWD procedures had a recurrence rate of 7% (29 studies; 95% CI: 4–9%; I^2^ = 94.43%). Within this group, CWDM showed 6% recurrence (8 studies; 95% CI: 2–10%; I^2^ = 72.16%), CWDT had 7% recurrence (7 studies; 95% CI: 0–13%; I^2^ = 98.23%), and CWDTM had the lowest recurrence at 1% (3 studies; 95% CI: 0–2%) with almost no heterogeneity (I^2^ = 0.03%).

CWU procedures had a higher recurrence rate of 16% (27 studies; 95% CI: 11–21%; I^2^ = 95.78%). Subgroup analysis showed CWUM at 11% (4 studies; 95% CI: 1–16%; I^2^ = 40.40%), CWUT at 19% (2 studies; 95% CI: 10–27%; I^2^ = 47.90%), and CWUTM at 19% (2 studies; 95% CI: 10–27%; I^2^ = 47.90%).

Other surgical approaches showed the following recurrence rates: combined approach tympanoplasty 17% (2 studies; 95% CI: 0–39%; I^2^ = 92.45%), endoscopic approach 8% (14 studies; 95% CI: 5–11%; I^2^ = 79.81%), TEA 8% (10 studies; 95% CI: 4–11%; I^2^ = 83.60%), mastoid-sparing surgery 11% (4 studies; 95% CI: 3–19%; I^2^ = 92.32%), microscopic approach 11% (2 studies; 95% CI: 0–26%; I^2^ = 80.67%), retrograde mastoidectomy 8% (3 studies; 95% CI: 1–14%; I^2^ = 56.35%), and iCW procedure 12% (2 studies; 95% CI: 7–17%; I^2^ = 85.39%).

#### 3.4.6. Mastoid Obliteration, Ossicular Reconstruction, and Perioperative Ventilation Tube-Based Rates

Studies that performed mastoid obliteration showed a lower recurrence rate of 9% (14 studies; 95% CI: 4–14%; I^2^ = 96.66%) compared to no mastoid obliteration, where recurrence was 29% (4 studies; 95% CI: 11–47%; I^2^ = 94.63%) (Figure 4).

Studies reporting complete ossicular reconstruction showed a recurrence rate of 6% (12 studies; 95% CI: 3–9%; I^2^ = 62.37%). In contrast, those reporting no reconstruction had a higher recurrence rate of 16% (3 studies; 95% CI: 4–28%; I^2^ = 54.15%) (Figure 5).

The presence of a perioperative ventilation tube was associated with a recurrence rate of 18% (2 studies; 95% CI: 0–36%; I^2^ = 90.6%; low certainty), while its absence corresponded to a higher recurrence rate of 26% (2 studies; 95% CI: 0–59%; I^2^ = 92.79%) (Appendix A).

#### 3.4.7. Follow-Up Based Recurrence Rate

The pooled recurrence rate varied across different follow-up durations (Appendix A). At 12 months, the recurrence rate was 7% (9 studies; 95% CI: 3–12%; I^2^ = 96.76%), while at 24 months, it was slightly higher at 9% (16 studies; 95% CI: 6–13%; I^2^ = 95.76%). A notable increase in recurrence was observed at 36 months, reaching 16% (16 studies; 95% CI: 11–21%; I^2^ = 91.06%). This trend continued at 48 months with 15% (13 studies; 95% CI: 9–21%; I^2^ = 94.97%) and peaked at 60 months with 18% (23 studies; 95% CI: 12–24%; I^2^ = 98.14%).

For long-term follow-ups, the recurrence rate remained high, with 120 months showing 17% (11 studies; 95% CI: 8–27%; I^2^ = 98.85%). Beyond this point, recurrence rates increased significantly. At 144 months, the recurrence rate was 20% (2 studies; 95% CI: 10–31%; I^2^ = 73.26%), while at 180 months, it reached 39% (2 studies; 95% CI: 21–56%; I^2^ = 96.58%). Beyond this point, very few studies with substantial dropout and high heterogeneity contributed data; therefore, we excluded those estimates from the final analysis. Long-term recurrence beyond 15 years should thus be regarded as exploratory and warrants future investigation through prospective time-to-event analyses.

### 3.5. Meta-Regression Findings

The meta-regression analysis examined the association between various surgical and patient-related factors with recurrence rates (Table 2). In the unadjusted model, several variables were found to have statistically significant associations with recurrence. Revision surgery demonstrated a strong positive association (β = 0.0032, *p* < 0.001). Similarly, planned second-look surgery was also significantly associated with recurrence (β = 0.0010, *p* = 0.004). Staged surgery, age, the definition of recurrence, follow-up duration, ossicular reconstruction, ossicular erosion, mastoid obliteration, and acquired cholesteatoma all exhibited statistically significant effects in the unadjusted model.

In the adjusted model, revision surgery lost statistical significance (*p* = 0.485). Interestingly, planned second-look surgery demonstrated an inverse association with recurrence (β = −0.0145, *p* = 0.030). Staged surgery remained significantly associated with recurrence (β = 0.0153, *p* = 0.029). Age also remained an independent predictor of recurrence (β = 0.0036, *p* = 0.006).

## 4. Discussion

Drawing from 84 studies, we observed that the recurrence rate of cholesteatoma was considerably variable driven by patient age, disease extent, surgical approach, and follow-up duration. These findings not only reinforce established notions but also challenge and refine prevailing surgical paradigms.

### 4.1. Geographic Variation and Demographics

The marked variation in recurrence rates across countries—ranging from 3% in China and England to 29% in India—likely reflects a complex interplay of socioeconomic status, healthcare infrastructure, surgeon experience, and patient follow-up adherence. The marked variation in recurrence rates across countries—ranging from 3% in China and England to 29% in India—likely reflects a complex interplay of healthcare infrastructure, availability of long-term follow-up, surgical case volume, and reporting practices. Our dataset did not provide details on surgeon training or institutional resources, and it would be speculative to attribute differences to the level of surgical training alone. Instead, these findings underscore the need for standardized reporting frameworks and multicenter prospective data that can disentangle healthcare system-related influences from patient or disease-specific factors.

These disparities echo earlier findings from Tomlin et al. [10], who emphasized that recurrence risk is partially modulated by regional surgical norms and access to second-look procedures. Similarly, the higher recurrence in children compared to adults aligns with Shewel et al. [102], who attributed the elevated risk to anatomical immaturity, aggressive epithelial proliferation, and eustachian tube dysfunction in the pediatric population.

Interestingly, recurrence did not differ significantly by gender or ear laterality, although slight heterogeneity in these analyses suggests further exploration may be warranted in future prospective studies.

### 4.2. Disease Characteristics and Staging

Two of the most clinically important findings from this review are the higher recurrence in children versus adults and the strong stage-dependent gradient in recurrence risk. Children consistently demonstrated higher recurrence than adults (13% vs. 10%), while advanced-stage disease was associated with progressively greater recurrence—from 4% in Stage I to 18% in Stage IV. These results emphasize that both age group and disease stage are critical prognostic factors that should directly influence surgical decision-making and long-term monitoring.

Consistent with the EAONO/JOS classification, our results demonstrate a clear stage-dependent increase in recurrence, from 4% in Stage I to 18% in Stage IV disease. This gradient underscores the clinical value of accurate staging for prognostication and surgical planning. However, our findings contrast with Körmendy et al. [103], who found no significant predictive utility of the Potsic staging system for congenital cholesteatoma, highlighting the limited translatability of congenital classification systems to acquired cases. Moreover, acquired cholesteatoma was associated with a higher recurrence than congenital types, a finding previously hypothesized by van der Toom [11], who linked acquired disease with underlying eustachian tube pathology, retraction pockets, and delayed diagnosis.

### 4.3. Surgical Technique and Intent

Among the most clinically actionable findings is the lower recurrence associated with CWD techniques compared to CWU approaches. These results mirror those of Shewel et al. [102] and Tomlin et al. [10], both of whom reported significantly higher recidivism in CWU procedures, particularly in children. Subclassification within CWU and CWD procedures further clarified this trend, with CWUT and CWUTM subtypes exhibiting recurrence rates exceeding 19%, while CWDTM—an ossiculoplasty-preserving variant of CWD—had the lowest recurrence at only 1%.

Our data also reinforce the value of adjunct techniques. Mastoid obliteration was associated with a reduced recurrence (9%) versus non-obliteration techniques (29%), consistent with van der Toom’s multi-institutional findings supporting obliterative surgery in both primary and revision cases [11]. Similarly, complete ossicular reconstruction appears protective, corroborating Li et al.’s meta-analytic findings that ossiculoplasty plays a dual role in hearing restoration and disease control [10,104].

Interestingly, while single-stage and staged surgeries had comparable recurrence, the adjusted meta-regression suggested that staged surgery independently predicts recurrence risk. This may reflect selection bias, where more extensive disease necessitates staged approaches. Nonetheless, the adjusted inverse association between planned second-look surgery and recurrence suggests that a proactive surgical strategy may help mitigate long-term risk—an insight supported by Li et al. [104] and Amoodi et al. [8], who advocated for individualized second-look planning aided by non-EPI diffusion-weighted MRI (DWI).

### 4.4. Follow-Up Duration and Recurrence Trends

Our analysis demonstrates a clear time-dependent increase in recurrence, reaching nearly 40% at 15–25 years postoperatively. This underlines a crucial message: shorter follow-up windows—typically 12 to 24 months—may drastically underestimate long-term disease burden. The cumulative pattern we observed calls for standardized long-term surveillance protocols. Our findings resonate with van der Toom [11], who emphasized the necessity of extending radiological or clinical follow-up beyond five years to capture true recurrence patterns.

It should also be emphasized that heterogeneity was very high across many subgroup analyses (I^2^ often >90%). Although meta-regression (Table 2) identified several contributing factors, residual heterogeneity remains substantial. Accordingly, small percentage differences between subgroups should be interpreted cautiously, as they may not be clinically meaningful under such variability.

### 4.5. Role of Imaging in Recurrence Detection

Although not the primary focus of this study, our findings indirectly support the growing role of non-EPI DWI in recurrence detection. Several recent meta-analyses, including those by Xun et al. [105], Amoodi et al. [8], and Muzaffar et al. [106], have consistently reported sensitivities and specificities >90% for non-EPI DWI in detecting residual and recurrent cholesteatoma. These data suggest that integrating advanced imaging into follow-up protocols—particularly in CWU procedures—may obviate unnecessary second-look surgeries while enabling early intervention in confirmed cases.

### 4.6. Limitations

This review has several limitations that should be considered when interpreting the pooled estimates. First, statistical heterogeneity was high across many subgroup analyses (I^2^ often >90%), reflecting inconsistent recurrence definitions, varied surgical techniques, and wide follow-up ranges. Although we used subgroup analyses and meta-regression, residual confounding remains likely; consequently, small percentage differences should be interpreted cautiously rather than as clinically decisive effects. Furthermore, we did not directly analyze residual disease versus true recurrence, a distinction emphasized by both Muzaffar [106] and Amoodi et al. [8], as imaging advancements now allow more precise differentiation.

Second, outcome definition and detection were inconsistent. In routine practice, the boundary between residual and recurrent cholesteatoma is often blurred, and many included studies did not clearly distinguish the two, which may inflate heterogeneity. Surveillance modality (e.g., routine non-EPI DWI versus clinical/otoscopic follow-up) was rarely and inconsistently reported; very few studies stratified recurrence by detection method, precluding a meaningful subgroup analysis. Standardized definitions and surveillance reporting would substantially improve comparability.

Third, key anatomic predictors were underreported. Potentially important risk factors—lateral semicircular canal (LSC) dehiscence/erosion, facial nerve canal dehiscence, and tegmen defects—were inconsistently documented, and factor-specific numerators/denominators were seldom provided, preventing meta-analytic estimation. Future work should adopt standardized radiologic/intraoperative descriptors to enable quantitative synthesis.

Fourth, interpretation of planned second-look surgery is limited by confounding by indication. Crude pooling suggested higher recurrence where a second look was planned, whereas the adjusted meta-regression indicated an inverse association. Surgeons are more likely to schedule second-look procedures for extensive or higher-risk disease, biasing crude estimates upward. Given inconsistent covariate reporting and because additional sensitivity analyses (e.g., Hartung–Knapp, leave-one-out) were not prespecified, residual confounding cannot be excluded.

Finally, very long-term follow-up estimates are sparse. Data beyond 15 years came from few studies with substantial dropout and heterogeneity; therefore, we restricted cumulative reporting to ≤15 years and consider longer-term figures exploratory.

## 5. Conclusions

This systematic review and meta-analysis provide a robust synthesis of the global evidence on cholesteatoma recurrence, offering critical insights into the epidemiological, clinical, and surgical determinants of disease relapse. Our findings highlight the substantial heterogeneity in recurrence rates, influenced by geographic variation, age, cholesteatoma type and stage, and surgical technique. Canal wall down procedures, mastoid obliteration, complete ossicular reconstruction, and planned second-look surgeries appear to confer lower recurrence risks, while longer follow-up durations reveal a significantly higher cumulative disease burden than previously appreciated. To optimize patient outcomes and resource allocation, future research should focus on prospective, multicenter trials using standardized definitions and incorporating validated staging systems.

## Figures and Tables

**Figure 1 biomedicines-13-02506-f001:**
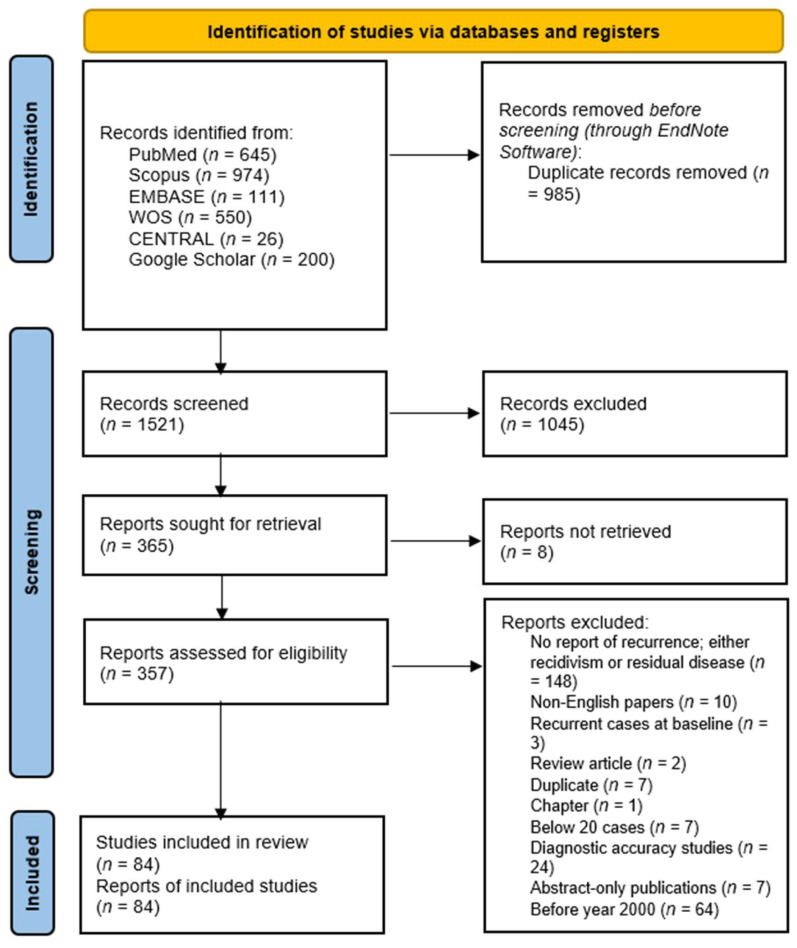
A PRISMA flowchart showing the results of the database search.

**Figure 2 biomedicines-13-02506-f002:**
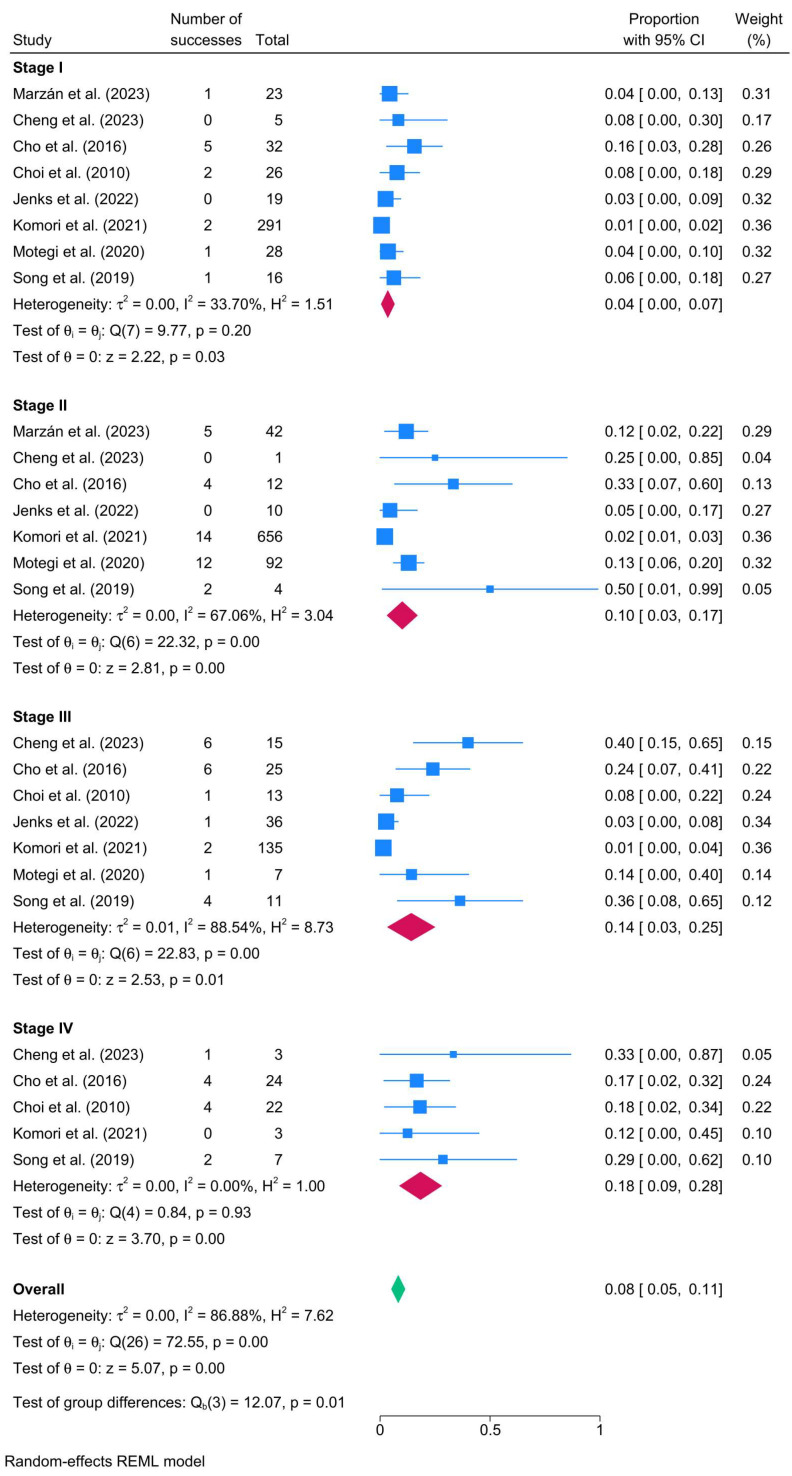
Forest plot showing the pooled cholesteatoma recurrence rate stratified by cholesteatoma stage [24,29,30,31,53,57,70,88].

**Figure 3 biomedicines-13-02506-f003:**
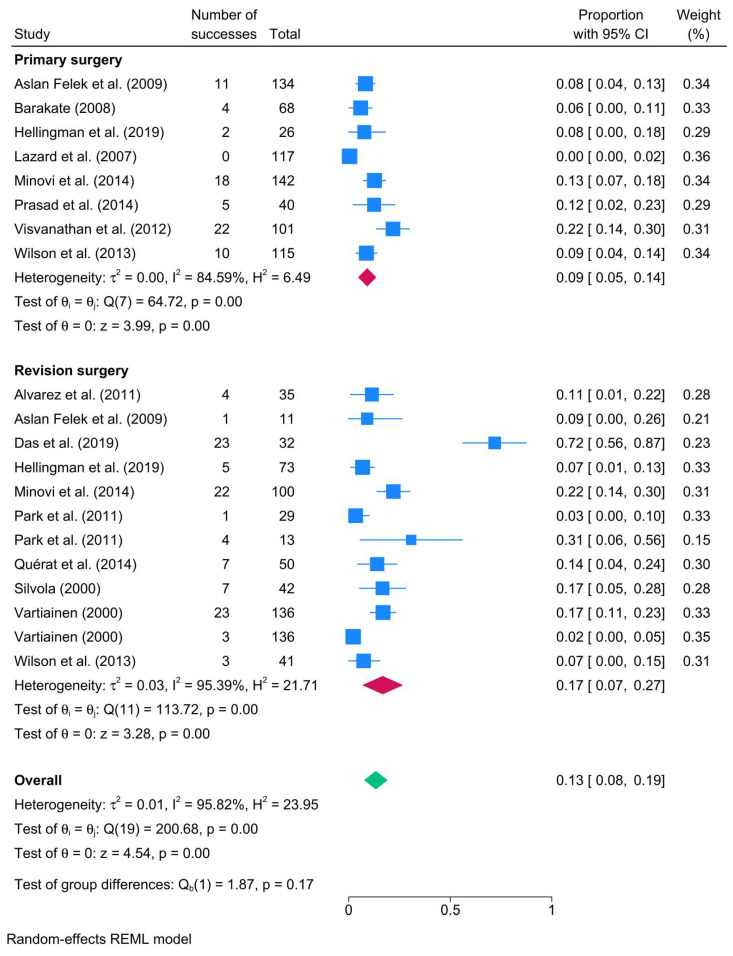
Forest plot showing the pooled cholesteatoma recurrence rate stratified by surgical indication (primary vs. revision surgery) [23,25,27,35,47,59,64,76,78,81,87,93,94,96].

**Figure 4 biomedicines-13-02506-f004:**
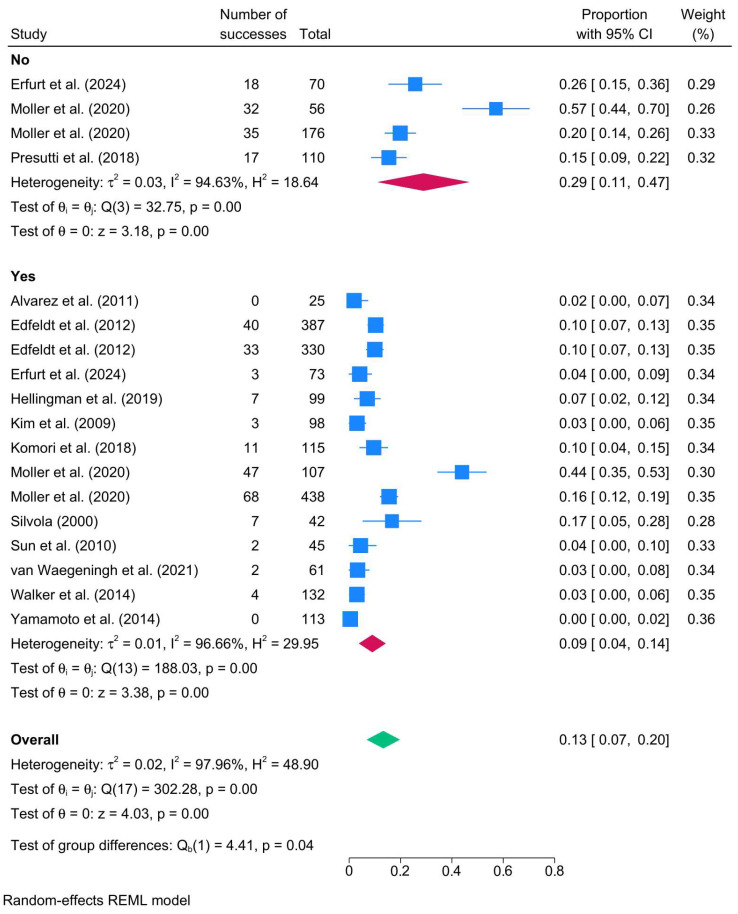
Forest plot showing the pooled cholesteatoma recurrence rate stratified by mastoid obliteration (yes vs. no) [23,39,40,41,47,55,56,67,79,87,89,92,95,98].

**Figure 5 biomedicines-13-02506-f005:**
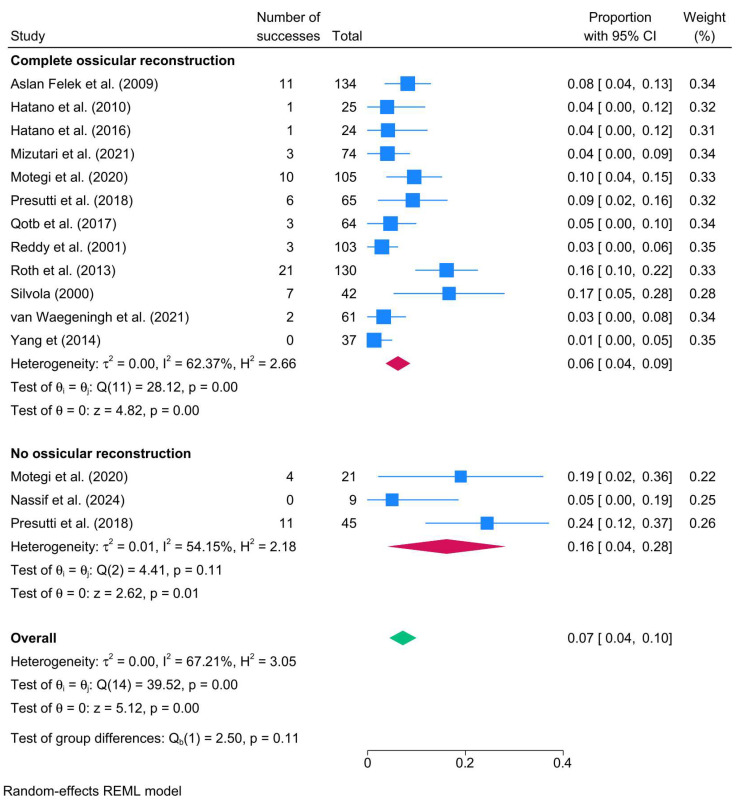
Forest plot showing the pooled cholesteatoma recurrence rate stratified by ossicular reconstruction (yes vs. no) [25,45,46,66,70,72,79,80,82,83,87,92,99].

**Table 1 biomedicines-13-02506-t001:** A summary of the meta-analytic estimates of cholesteatoma recurrence across all examined subgroups.

	Number of Studies	Pooled Rate (%)	95% CI	I^2^ (%)	*p*-Value
Country					0.001
Belgium	1	3%	0–8%	N/A	
China	4	3%	1–6%	0.05%	
Czech Republic	1	7%	0–19%	N/A	
Denmark	3	24%	0–48%	98.91%	
Egypt	1	5%	0–10%	N/A	
England	1	3%	0–6%	N/A	
Finland	2	17%	11–22%	0.01%	
France	4	6%	0–12%	88.14%	
Germany	2	12%	9–15%	0.12%	
India	3	29%	0–70%	98.61%	
Israel	2	24%	15–34%	10.88%	
Italy	12	12%	7–17%	96.44%	
Japan	12	7%	4–10%	95.16%	
Korea	8	8%	3–13%	79.81%	
Malaysia	1	4%	0–10%	N/A	
Senegal	1	14%	5–22%	N/A	
Spain	2	10%	4–16%	0%	
Sweden	2	10%	8–12%	0.02%	
Switzerland	1	16%	10–22%	N/A	
Taiwan	2	4%	0–9%	29.49%	
The Netherlands	4	13%	3–23%	90.51%	
Turkey	2	12%	5–20%	85.20%	
United Kingdom	5	7%	0–14%	91.51%	
United States of America	8	6%	3–9%	75.37%	
Age					0.001
Adults	25	10%	6–14%	97.80%	
Children	36	13%	9–16%	95.33%	
Adults and Children	15	11%	4–18%	97.19%	
Gender					0.96
Male	3	17%	8–27%	76.09%	
Female	3	18%	10–25%	24.20%	
Laterality					0.79
Right	3	15%	6–25%	31.87%	
Left	3	18%	3–32%	77.29%	
Cholesteatoma Type					0.2
Acquired	14	12%	8–16%	80.35%	
Congenital	13	7%	3–11%	92.36%	
Recurrent Cholesteatoma Definition					0.62
Defined	31	11%	7–14%	96.61%	
Not Defined	53	10%	7–12%	97.64%	
Stage of Cholesteatoma					0.01
Stage I	8	4%	0–7%	33.70%	
Stage II	7	10%	3–17%	67.06%	
Stage III	7	14%	3–25%	88.54%	
Stage IV	5	18%	9–28%	0%	
Cholesteatoma Location					0.001
Attic	11	10%	4–15%	91.06%	
Aural	2	20%	12–28%	86.74%	
Mastoid	3	16%	3–30%	72.51%	
Middle Ear (not classified)	7	16%	9–23%	86.57%	
Pars Flaccida	3	12%	0–25%	95.37%	
Pars Tensa	3	8%	2–14%	85.56%	
Petrous bone	2	19%	5–33%	74.96%	
Pars Tensa + Flaccida	3	15%	2–28%	88.06%	
Tympano-mastoid	2	1%	0–2%	14.48%	
Surgical Intent					0.17
Primary Surgery	8	9%	5–14%	84.59%	
Revision Surgery	12	17%	7–27%	95.39%	
Staged Surgery					0.87
Single-stage surgery	11	8%	4–13%	91.41%	
Staged surgery (2 stages or more)	5	9%	6–12%	0.05%	
Surgery Type					0.001
Atticotomy-based procedure	4	7%	0–15%	91.33%	
CWD Procedure	29	7%	4–9%	94.43%	
CWDM	8	6%	2–10%	72.16%	
CWDT	7	7%	0–13%	98.23%	
CWDTM	3	1%	0–2%	0.03%	
CWU Procedure	27	16%	11–21%	95.78%	
CWUM	4	9%	1–16%	40.40%	
CWUT	2	19%	10–27%	47.90%	
CWUTM	9	12%	5–19%	92.45%	
Combined approach tympanoplasty	2	17%	0–39%	92.39%	
Endoscopic Approach	14	8%	5–11%	79.81%	
TEA	10	8%	4–11%	83.60%	
Mastoid-sparing surgery	4	11%	3–19%	92.32%	
Microscopic approach	2	11%	0–26%	80.67%	
Retrograde mastoidectomy	3	8%	1–14%	56.35%	
iCW Procedure	8	12%	7–17%	85.39%	
Second-look Surgery					0.42
Planned 2nd look	4	13%	6–19%	60.80%	
Not planned	5	10%	7–12%	32.64%	
Mastoid Obliteration					0.04
Yes	14	9%	4–14%	96.66%	
No	4	29%	11–47%	94.63%	
Ossicular Reconstruction (any)					0.11
Complete reconstruction	12	6%	4–9%	62.37%	
No reconstruction	3	16%	4–28%	54.15%	
Perioperative Ventilation Tube					0.64
Yes	2	18%	6–30%	0%	
No	2	26%	0–59%	92.79%	
Follow-up					0.001
12 months	9	7%	3–12%	96.76%	
24 months	16	9%	6–13%	95.76%	
30 months	3	7%	0–15%	77.74%	
36 months	16	16%	11–21%	91.06%	
48 months	13	15%	9–21%	94.97%	
55 months	2	9%	0–21%	77.34%	
60 months	23	18%	12–24%	98.14%	
72 months	6	10%	6–14%	81.35%	
84 months	3	14%	2–26%	82.94%	
96 months	3	15%	2–29%	87.86%	
120 months	11	17%	8–27%	98.85%	
132 months	2	14%	8–20%	57.90%	
144 months	2	20%	10–31%	73.26%	
180 months	2	39%	21–56%	96.58%	

I^2^: a measure of statistical heterogeneity; CI: confidence interval; CWU: canal wall-up; CWD: canal wall-down; CWUM: canal wall-up mastoidectomy; CWUT: canal wall-up tympanoplasty; CWUTM: canal wall-up tympanomastoidectomy; CWDM: canal wall-down mastoidectomy; CWDT: canal wall-down tympanoplasty; CWDTM: canal wall-down tympanomastoidectomy; CWR: canal wall reconstruction; EES: endoscopic ear surgery; T: tympanoplasty; CAT: combined approach tympanoplasty; TM: tympanomastoidectomy; M: mastoidectomy; iCWT: intact canal wall tympanoplasty; TEA: transcanal endoscopic approach; MO: mastoid obliteration; N/A = Not Applicable.

**Table 2 biomedicines-13-02506-t002:** Summary of the unadjusted and adjusted meta-regression analysis of the determinants of recurrence rate of cholesteatoma post-surgical management.

	Unadjusted Model						Adjusted Model					
	Coefficient	SE	Z	*p*-Value	Low CI	High CI	Coefficient	SE	Z	*p*-Value	Low CI	High CI
Revision surgery (per %)	0.0032	0.0004	7.49	<0.001	0.0024	0.0040	0.0016	0.0023	0.7	0.4850	−0.0029	0.0061
Planned second look (per %)	0.0010	0.0003	2.89	0.0040	0.0003	0.0016	−0.0145	0.0067	−2.16	0.0300	−0.0276	−0.0014
Staged surgery (per %)	0.0011	0.0004	3.06	0.0020	0.0004	0.0018	0.0153	0.0070	2.18	0.0290	0.0015	0.0291
Age (per year)	0.0028	0.0005	5.99	<0.001	0.0019	0.0037	0.0036	0.0013	2.76	0.0060	0.0010	0.0062
Recurrence defined (vs. not defined)	0.1100	0.0222	4.96	<0.001	0.0666	0.1534	−0.1215	0.0876	−1.39	0.1650	−0.2933	0.0502
Follow-up (per month)	0.0011	0.0001	7.68	<0.001	0.0008	0.0014	0.0004	0.0005	0.81	0.4170	−0.0006	0.0014
Ossicular reconstruction (per %)	0.0012	0.0004	3.08	0.0020	0.0004	0.0019	Omitted (due to collinearity)					
Ossicular erosion (per %)	0.0021	0.0004	5.82	<0.001	0.0014	0.0028	Omitted (due to collinearity)					
Mastoid obliteration (per %)	0.0711	0.0191	3.72	<0.001	0.0336	0.1085	Omitted (due to collinearity)					
Acquired Cholesteatoma (vs. Congenital)	0.1229	0.0249	4.94	<0.001	0.0741	0.1717	Omitted (no observations)					

SE: standard error; CI: confidence interval.

## Data Availability

The analyzed dataset was driven from published data in the literature; however, the full dataset can be provided by the corresponding author upon reasonable request.

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
