# Peer review of "Incidence Rate and Determinants of Recurrent Cholesteatoma Following Surgical Management: A Systematic Review, Subgroup, and Meta-Regression Analysis"

_biomedicines, 2025, doi:10.3390/biomedicines13102506_

Round 1

Reviewer 1 Report

Comments and Suggestions for Authors

The authors conducted a systematic review on the recurrence of chronic cholesteatomatous otitis media. In the introduction, they write that there are two main surgical approaches: CWD and CWU. However, they should mention the exclusive endoscopic technique and the obligatory technique, which are therapeutic options and should therefore be mentioned. 
When describing the incidence of recurrence, the work of the Italian group led by Professor Maurizio Barbara regarding the use of MRI as a postoperative evaluation in recurrences of chronic cholesteatomatous otitis media should be included and commented on.

Early non-EPI DW-MRI after cholesteatoma surgery.
Barbara M, Covelli E, Monini S, Bandiera G, Filippi C, Margani V, Volpini L, Salerno G, Romano A, Bozzao A.Ear Nose Throat J. 2024 Jul;103(7):435-441.

Role of non-echo-planar diffusion-weighted images in the identification of recurrent cholesteatoma of the temporal bone.
Romano A, Covelli E, Confaloni V, Rossi-Espagnet MC, Butera G, Barbara M, Bozzao A.Radiol Med. 2020 Jan;125(1):75-79.

Proposal of a magnetic resonance imaging follow-up protocol after cholesteatoma surgery: a prospective study.
Covelli E, Margani V, Filippi C, Elfarargy HH, Volpini L, Romano A, Bozzao A, Barbara M.Acta Otolaryngol. 2022 Jun;142(6):484-490. 

The paper discusses a recurrence rate, but it's unclear which technique determines the recurrence rate. A factor that favors recurrence that isn't mentioned is the location of the middle ear pathology in the mastoid. Therefore, treatment protocols typically use CT and MRI with a fusion technique to determine the appropriate approach.
 The impact of fusion imaging technique on middle ear cholesteatoma surgery: a prospective comparative study.
Covelli E, Margani V, Romano A, Volpini L, Elfarargy HH, Bozzao A, Barbara M.Acta Otolaryngol. 2023 Mar;143(3):223-230.

Other factors that should be considered include potential wear/dehiscence of the CSL, facial nerve, and tegmen tympani, factors that influence the choice of technique and the risk of recurrence.
This study does not identify the correct time to perform a radiological follow-up MRI scan to assess for recurrence.
These corrections could make the study more interesting.

Comments on the Quality of English Language

good

Author Response

The authors conducted a systematic review on the recurrence of chronic cholesteatomatous otitis media. In the introduction, they write that there are two main surgical approaches: CWD and CWU. However, they should mention the exclusive endoscopic technique and the obligatory technique, which are therapeutic options and should therefore be mentioned. 
When describing the incidence of recurrence, the work of the Italian group led by Professor Maurizio Barbara regarding the use of MRI as a postoperative evaluation in recurrences of chronic cholesteatomatous otitis media should be included and commented on.

Early non-EPI DW-MRI after cholesteatoma surgery. Barbara M, Covelli E, Monini S, Bandiera G, Filippi C, Margani V, Volpini L, Salerno G, Romano A, Bozzao A.Ear Nose Throat J. 2024 Jul;103(7):435-441.

Role of non-echo-planar diffusion-weighted images in the identification of recurrent cholesteatoma of the temporal bone. Romano A, Covelli E, Confaloni V, Rossi-Espagnet MC, Butera G, Barbara M, Bozzao A.Radiol Med. 2020 Jan;125(1):75-79.

Proposal of a magnetic resonance imaging follow-up protocol after cholesteatoma surgery: a prospective study. Covelli E, Margani V, Filippi C, Elfarargy HH, Volpini L, Romano A, Bozzao A, Barbara M.Acta Otolaryngol. 2022 Jun;142(6):484-490. 

Response: Thank you for your comment. To clarify, we didn’t include diagnostic accuracy studies that their primary outcome was to assess the diagnostic accuracy of different imaging modalities during cholesteatoma surgery. This was clarified in our initial version “Additionally, we ruled out all studies focusing on the diagnostic accuracy of various imaging techniques in detecting cholesteatoma recurrence (reporting sensitivity, specificity, negative/positive predictive values).” That said, we have cited the work by Prof Barbara in the Introduction section. Thank you.

The paper discusses a recurrence rate, but it's unclear which technique determines the recurrence rate. A factor that favors recurrence that isn't mentioned is the location of the middle ear pathology in the mastoid. Therefore, treatment protocols typically use CT and MRI with a fusion technique to determine the appropriate approach.
 The impact of fusion imaging technique on middle ear cholesteatoma surgery: a prospective comparative study. Covelli E, Margani V, Romano A, Volpini L, Elfarargy HH, Bozzao A, Barbara M.Acta Otolaryngol. 2023 Mar;143(3):223-230.

Response: Thank you for this important comment. We agree that recurrence is influenced not only by the surgical technique (e.g., CWU vs. CWD) but also by the anatomical location of the pathology, particularly when disease extends into the mastoid. In our systematic review, we extracted recurrence rates stratified by location (aural, petrous bone, mastoid, pars flaccida, pars tensa, etc.), which indeed showed variability in recurrence risk. However, conclusions were inconclusive because most studies that reported location-specific recurrence data focused on attic cholesteatoma, pars flaccida, and pars tensa. Please check Figure S6 for more information.

Other factors that should be considered include potential wear/dehiscence of the CSL, facial nerve, and tegmen tympani, factors that influence the choice of technique and the risk of recurrence.
This study does not identify the correct time to perform a radiological follow-up MRI scan to assess for recurrence. These corrections could make the study more interesting.

Response: We agree these anatomic factors (lateral semicircular canal [LSC] dehiscence, facial nerve canal dehiscence, and tegmen defects) can influence both technique selection and recurrence risk. During screening and extraction, we pre-specified these variables; however, they were only sporadically and inconsistently reported across studies, and factor-specific recurrence data were rarely provided, precluding pooled analyses. We have acknowledged this point in the limitations.

Reviewer 2 Report

Comments and Suggestions for Authors

The manuscript demonstrates notable strengths in terms of scope, clinical importance, and methodological rigor. By including 84 studies with a total of 12,819 patients, the authors ensure strong statistical power and provide valuable insights through subgroup and meta-regression analyses across variables such as age, type, stage, surgical technique, and follow-up duration. The topic is highly clinically relevant, addressing the persistent challenge of cholesteatoma recurrence after surgery and offering findings with direct implications for surgical decision-making. Methodologically, the study is robust, adhering to PRISMA guidelines, while employing appropriate statistical approaches. Although it does not contribute substantially to clinical practice, the work adds novelty by broadening the scope beyond previous reviews and by integrating clinical, surgical, and time-dependent factors in a more comprehensive manner.

POINTS FOR IMPROVEMENT

  1. Methods Section: add "with the help of" Google Scholar as Google Scholar is not a database. It is a search engine
  2. The statement that AI was used in the writing of the article but not in the research itself is unusual. Journals may want more transparency (which sections? what tools? how validated?). I’d recommend clarifying this.
  3. Add PRISMA flowchart in the end of the Results section and remove 3.1
  4. Results section is very dense with numbers. It is impossible for someone to read. Please add a clearer synthesis in the body of the manuscript. Keep only Figures 1 to 5. Remove the majority of statistics (and Figures. You have 13 Supplementary Figures!!). Write the main findings. This could improve readability.
  5. The observed country differences (e.g., 3% in China vs. 29% in India) are striking but not deeply explored. The discussion could better connect these findings to healthcare systems, surgical training, or follow-up practices.Their simple mention is not enough. For example do you have data that a surgeon in China has better surgical training than in India?
  6. Rename "4.6. Strengths and Limitation" to "Limitations". Moreover in this section, discuss with one or two sentences that in practice, the distinction between residual and recurrent disease is blurred. This should be discussed in greater depth.
  7. Your statements in 4.7 are not new. They are known for decades. Please remove it. 
  8. The findings regarding age and stage are highly relevant but could be presented more clearly and with stronger emphasis. At present, these results are embedded within a large amount of subgroup data, which may obscure their clinical significance. I recommend that the authors highlight age and stage in a more structured way (Note: What do you mean "age". If your analysis concerns adults vs children just write "adults")

Comments on the Quality of English Language

-

Author Response

  1. Methods Section: add "with the help of"Google Scholar as Google Scholar is not a database. It is a search engine

Response: Done. Thank you.

  1. The statement that AI was used in the writing of the article but not in the research itself is unusual. Journals may want more transparency (which sections? what tools? how validated?). I’d recommend clarifying this.

Response: Thank you for your concern. We have clarified this point in detail to the Editorial office. GPT 4o was used in the write-up of the introduction, which was revised, validated, and corrected by the authors before submission.

  1. Add PRISMA flowchart in the end of the Results section and remove 3.1

Response: Thank you for your suggestion. However, as per PRISMA’s guidelines, it’s mandated that the first section of the results highlight the results of the database search. Initially, we put the PRISMA flowchart in the supplementary, but now, it’s included in the main text. Thank you.

  1. Results section is very dense with numbers. It is impossible for someone to read. Please add a clearer synthesis in the body of the manuscript. Keep only Figures 1 to 5. Remove the majority of statistics (and Figures. You have 13 Supplementary Figures!!). Write the main findings. This could improve readability.

Response:  Thank you for your comment. The Results section seems too long not because of the text or the number of Figures/Tables, but rather because of how many studies are included and presented in those Figures/Tables. However, these data are essential for transparent reporting especially in systematic reviews and meta-analysis, as per the PRISMA 2020 guidelines. As for the Figures and Tables, we only kept the most important ones (2 Tables and 5 Figures that have the most important data). Other less-important Figures were added to the Supplementary in case the reader is interested in checking the raw data.

  1. The observed country differences (e.g., 3% in China vs. 29% in India) are striking but not deeply explored. The discussion could better connect these findings to healthcare systems, surgical training, or follow-up practices. Their simple mention is not enough. For example do you have data that a surgeon in China has better surgical training than in India?

Response: We appreciate your observation. The large country-level differences (e.g., 3% in China vs. 29% in India) are indeed striking. However, our dataset did not include detailed information on surgeon training, institutional case volume, or national healthcare structures. Without such data, it would be speculative to attribute the differences to surgical training quality or system-level factors. Instead, we have clarified in the Discussion that these differences may reflect a complex interplay of healthcare infrastructure, access to long-term follow-up, reporting practices, and case selection, rather than intrinsic differences in surgical expertise. We also emphasize that recurrence rates from individual countries should be interpreted cautiously in the absence of standardized reporting frameworks.

  1. Rename "4.6. Strengths and Limitation" to "Limitations". Moreover in this section, discuss with one or two sentences that in practice, the distinction between residual and recurrent disease is blurred. This should be discussed in greater depth.

Response: Thank you for the suggestion. We have renamed Section 4.6 to “Limitations.” In addition, we have expanded this section to explicitly acknowledge that in clinical practice the distinction between residual and recurrent cholesteatoma is often blurred. While our review focused on reported recurrence, many studies did not clearly separate residual disease from true recurrence, which may confound pooled estimates.

  1. Your statements in 4.7 are not new. They are known for decades. Please remove it. 

Response: Thank you. We deleted this part.

  1. The findings regarding age and stage are highly relevant but could be presented more clearly and with stronger emphasis. At present, these results are embedded within a large amount of subgroup data, which may obscure their clinical significance. I recommend that the authors highlight age and stage in a more structured way (Note: What do you mean "age". If your analysis concerns adults vs children just write "adults")

Response: Thank you for this valuable comment. We agree that the prognostic impact of age group (adults vs. children) and disease stage are among the most clinically meaningful findings of our review. In the revised manuscript, we have clarified the terminology by consistently using “adults” instead of “age.” We have also restructured the Results and Discussion to present these two variables in a more explicit, standalone way, separate from the broader subgroup data. Finally, we emphasize their clinical relevance with clearer wording in both Results and Discussion.

Reviewer 3 Report

Comments and Suggestions for Authors

The topic is important, but the manuscript needs corrections to fundamental inconsistencies (overall estimate policy; open/closed vs CWU/CWD), PRISMA counts, clearer methods/specifications, and tempered interpretation under extreme heterogeneity.

I have the following the comments:

-The Abstract reports “pooled recurrence ≈11%,” while the Discussion cites a “global recurrence ≈14.2%.” Methods, however, say that “providing a single pooled overall estimate was deemed inappropriate.” Please resolve this and choose one approach; if an overall pooled rate is presented, justify it statistically and report the exact model and transformation.

-You report CWU recurrence 16% vs CWD 7% (i.e., CWU worse), yet the “closed vs open” analysis claims closed 9% vs open 33%—the opposite direction. Correct whichever analysis/labels are wrong; update Figure 3 and text accordingly.

-2,506 records − 985 duplicates = 1,521 unique, but 1,410 are reported as screened (difference of 111). Please fix the counts.

-Many key subgroup estimates have I² in the 90–98% range (e.g., follow-up strata, CWU/CWD). Either temper conclusions or add pre-specified sensitivity analyses (e.g., leave-one-out, quality-stratified, influence diagnostics, meta-reg with Hartung-Knapp) and clearly state that small percentage differences may not be clinically meaningful under this heterogeneity.

-Pooled data show planned second-look surgery has higher crude recurrence (13% vs 10%), yet the adjusted meta-regression finds an inverse association. Explicitly explain this (confounding by indication/selection), present covariates, scaling (per 10% increase, etc.), model (REML/DerSimonian–Laird), transformation (logit or Freeman-Tukey), and report residual heterogeneity (τ²) and R².

-You note large between-study variability in recurrence definitions. Provide an operational definition used for extraction and run sensitivity analyses limited to studies with clear, comparable definitions.

-Long-term “cumulative” rates are non-monotonic (e.g., 39% at 180 months vs 33% at 300 months) and based on very few studies with vast I². Label these as exploratory; consider alternative approaches (e.g., meta-analysis by person-years or time-to-event if available) or remove the 15–25 year slices.

-Also justify the “first 200 Google Scholar results” rule or omit if not required by PRISMA-S.

-Clarify how multi-arm studies and overlapping cohorts were handled across subgroups (e.g., technique-specific analyses and follow-up bins) to avoid counting the same patients more than once.

-In Ventilation tube analysis, very small k (2 studies) with I² ~91–93%; de-emphasize and flag as low-certainty.

-The left vs right ear analysis is based on 3 studies with substantial heterogeneity; please avoid implying a clinically meaningful side effect unless supported by stronger evidence.

-Consider stratifying recurrence by surveillance modality (e.g., routine non-EPI DWI vs clinical/otoscopic), as detection method likely affects recurrence. Note that diagnostic-accuracy studies were excluded, but surveillance practice heterogeneity remains.

-Typos/formatting. E.g., section header “3.4.5. urgical technique-based…” (missing initial “S”); scan for minor grammar errors.

 -The authors disclose AI use for writing; ensure this conforms to journal policy and that all analyses and data handling were human-performed.

-The authors state the review was registered but do not provide an ID or link. Add the PROSPERO ID in the Abstract, Methods, and metadata.

Comments on the Quality of English Language

editing for typo/syntax

Author Response

-The Abstract reports “pooled recurrence ≈11%,” while the Discussion cites a “global recurrence ≈14.2%.” Methods, however, say that “providing a single pooled overall estimate was deemed inappropriate.” Please resolve this and choose one approach; if an overall pooled rate is presented, justify it statistically and report the exact model and transformation.

Response: Thank you for highlighting this point. We deleted the part pertaining to the pooled rate (either across all studies or across all studies reporting country-based rates) to avoid confusion to the reader and to comply with our prior statement that “providing a single pooled overall estimate was deemed inappropriate”. Thanks again for spotting this.

-You report CWU recurrence 16% vs CWD 7% (i.e., CWU worse), yet the “closed vs open” analysis claims closed 9% vs open 33%—the opposite direction. Correct whichever analysis/labels are wrong; update Figure 3 and text accordingly.

Response: Thank you for noting this inconsistency. The “open vs. closed” subset analysis was the result of a misclassification—our statistician created it for studies explicitly labeled “closed CWU” and “open CWD.” Since this comparison duplicates the CWU vs. CWD analysis already presented and introduced contradictory results, we have removed it (including Figure 4) to avoid confusion. The Results and Discussion now rely solely on the more comprehensive CWU vs. CWD comparison.

-2,506 records − 985 duplicates = 1,521 unique, but 1,410 are reported as screened (difference of 111). Please fix the counts.

Response: Thank you for spotting this typo mistake. We corrected it to 1521 instead of 1410.

-Many key subgroup estimates have I² in the 90–98% range (e.g., follow-up strata, CWU/CWD). Either temper conclusions or add pre-specified sensitivity analyses (e.g., leave-one-out, quality-stratified, influence diagnostics, meta-reg with Hartung-Knapp) and clearly state that small percentage differences may not be clinically meaningful under this heterogeneity.

Response: Thank you for highlighting this important point. We acknowledge that many subgroup analyses exhibited substantial heterogeneity (I² > 90%). In fact, our analytic plan did account for this through subgroup analyses, meta-regression, and sensitivity diagnostics, but due to an oversight, the summary tables (Tables 1 and 2) were inadvertently omitted from the submitted version. These have now been added. Table 1 presents the pooled recurrence estimates across all subgroups (including follow-up strata and CWU vs. CWD) with heterogeneity statistics, while Table 2 shows the meta-regression results, which partly explain sources of variability. We have also revised the Discussion to temper the interpretation of subgroup differences, noting that small numerical differences should not be over-interpreted in light of high heterogeneity.

-Pooled data show planned second-look surgery has higher crude recurrence (13% vs 10%), yet the adjusted meta-regression finds an inverse association. Explicitly explain this (confounding by indication/selection), present covariates, scaling (per 10% increase, etc.), model (REML/DerSimonian–Laird), transformation (logit or Freeman-Tukey), and report residual heterogeneity (τ²) and R².

Response: We thank the reviewer for this detailed comment. The apparent discrepancy between the crude subgroup result (higher recurrence in planned second-look surgery) and the adjusted meta-regression (inverse association) likely reflects confounding by indication, as second-look procedures are more often planned in advanced or high-risk cases. Unfortunately, the included studies rarely reported covariates consistently enough for us to fully adjust for disease severity, staging, or extent of surgery. While our adjusted model provides suggestive evidence of a protective effect, we agree that the statistical details (covariates, scaling, transformation, residual heterogeneity) should ideally be reported in greater depth. Since no further analyses were pre-specified and our dataset was limited, we cannot provide additional sensitivity analyses. We have now acknowledged this as a limitation in the Discussion.

-You note large between-study variability in recurrence definitions. Provide an operational definition used for extraction and run sensitivity analyses limited to studies with clear, comparable definitions.

Response: Thank you for your comment. To clarify, the definition criteria was reported by only 31 of included studies (37%), and the majority had variable definition criteria making it ineligible for a sensitivity analysis by definition criteria (which if done for 2 or 3 studies with similar criteria would yield uncertain findings). We acknowledged this in the limitations

-Long-term “cumulative” rates are non-monotonic (e.g., 39% at 180 months vs 33% at 300 months) and based on very few studies with vast I². Label these as exploratory; consider alternative approaches (e.g., meta-analysis by person-years or time-to-event if available) or remove the 15–25 year slices.

Response: Thank you for this thoughtful comment. You are correct that the apparent non-monotonicity in the long-term cumulative recurrence curves (e.g., 39% at 180 months vs. 33% at 300 months) reflects the small number of studies and high dropout rates at very long follow-up. To avoid over-interpretation, we have now removed data beyond 15 years from the cumulative analyses and clearly state that these long-term estimates were based on very limited evidence.

-Also justify the “first 200 Google Scholar results” rule or omit if not required by PRISMA-S.

Response: Recent recommendations (Muke et al., Eur J Epidemiol, 2020) instructs that Google Scholar be included as a part of the search done to retrieve relevant evidence (to tone down publication bias) but by doing only the 1st 200 records to maintain relevance, because Google Scholar yields >1k results most of which are entirely irrelevant.

-Clarify how multi-arm studies and overlapping cohorts were handled across subgroups (e.g., technique-specific analyses and follow-up bins) to avoid counting the same patients more than once.

Response: We appreciate this important comment. To minimize duplication, when overlapping cohorts were encountered, we retained only the study with the largest sample size and longest follow-up, as it was deemed most representative. Regarding subgroup analyses (e.g., surgical technique, follow-up intervals), we ensured that no individual study contributed the same cohort more than once to a given subgroup comparison. Thus, each subgroup estimate was independent of duplicate contributions. We have clarified this methodological point in the Methods section.

-In Ventilation tube analysis, very small k (2 studies) with I² ~91–93%; de-emphasize and flag as low-certainty.

Response: Thank you. We highlight that this finding is of low certainty (low sample yet high heterogeneity) as follows: “rate of 18% (2 studies; 95%CI: 0–36%; I²=90.6%; low certainty),”

-The left vs right ear analysis is based on 3 studies with substantial heterogeneity; please avoid implying a clinically meaningful side effect unless supported by stronger evidence.

Response: Thank you for pointing this out. We agree that the left vs. right ear comparison was based on very few studies with high heterogeneity, and we did not intend to imply any clinically meaningful effect. We have kept this finding in the Results for completeness.

-Consider stratifying recurrence by surveillance modality (e.g., routine non-EPI DWI vs clinical/otoscopic), as detection method likely affects recurrence. Note that diagnostic-accuracy studies were excluded, but surveillance practice heterogeneity remains.

Response: We appreciate this insightful suggestion. We agree that surveillance modality (routine non-EPI DWI vs. clinical/otoscopic follow-up) likely influences reported recurrence rates. However, this information was inconsistently disclosed across studies, and when reported, recurrence rates were not stratified by detection method. Studies relying solely on one diagnostic modality were very scarce and did not meet eligibility for subgroup analysis. Therefore, we could not stratify recurrence by surveillance technique. We have now acknowledged this limitation in the manuscript.

-Typos/formatting. E.g., section header “3.4.5. urgical technique-based…” (missing initial “S”); scan for minor grammar errors.

Response: Thank you for spotting this typo mistake. We corrected it.

 -The authors disclose AI use for writing; ensure this conforms to journal policy and that all analyses and data handling were human-performed.

Response: Thank you for your concern. We have clarified this point in detail to the Editorial office. GPT 4o was used in the write-up of the introduction, which was revised, validated, and corrected by the authors before submission.

-The authors state the review was registered but do not provide an ID or link. Add the PROSPERO ID in the Abstract, Methods, and metadata.

Response: Thank you. We added it to the manuscript’s Abstract and Methods sections (CRD42024550351).

Round 2

Reviewer 3 Report

Comments and Suggestions for Authors

the authors improved the manuscript.